# VIDEO DIFFUSION MODELS

**Jonathan Ho,**[*] **Tim Salimans**[*]
{jonathanho,salimans}@google.com

**Alexey Gritsenko, William Chan, Mohammad Norouzi, David J. Fleet**
{agritsenko,williamchan,mnorouzi,davidfleet}@google.com

## ABSTRACT

Generating temporally coherent high fidelity video is an important milestone in generative modeling research. We make progress towards this milestone by proposing a diffusion model for video generation that shows very promising initial results. Our model is a natural extension of the standard image diffusion architecture, and it enables jointly training from image and video data, which we find to reduce the variance of minibatch gradients and speed up optimization. To generate long and higher resolution videos we introduce a new conditional sampling technique for spatial and temporal video extension that performs better than previously proposed methods. We present the first results on a large text-conditioned video generation task, as well as state-of-the-art results on an established unconditional video generation benchmark. Supplementary material is available at https://video-diffusion.github.io/.

## 1 INTRODUCTION

Diffusion models have recently been producing high quality results in image generation and audio generation (e.g. Kingma et al., 2021; Saharia et al., 2021a;b; Dhariwal & Nichol, 2021; Ho et al., 2021; Nichol et al., 2021; Song et al., 2021b; Whang et al., 2021; Salimans & Ho, 2021; Chen et al., 2021; Kong et al., 2021), and there is significant interest in validating diffusion models in new data modalities. In this work, we present first results on video generation using diffusion models, for both unconditional and conditional settings. Prior work on video generation has usually employed other types of generative models (e.g. Babaeizadeh et al., 2017; 2021; Lee et al., 2018; Kumar et al., 2019; Clark et al., 2019; Weissenborn et al., 2019; Yan et al., 2021; Walker et al., 2021).

We show that high quality videos can be generated using essentially the standard formulation of the Gaussian diffusion model (Sohl-Dickstein et al., 2015), with little modification other than straightforward architectural changes to accommodate video data within the memory constraints of deep learning accelerators. We train models that generate a fixed number of video frames, and enable generating longer videos by applying this model autoregressively using a new method for conditional generation. We test our methods on unconditional video generation, where we achieve state-of-the-art sample quality scores, and we also show promising first results on text-conditioned video generation.

## 2 BACKGROUND

A diffusion model (Sohl-Dickstein et al., 2015; Song & Ermon, 2019; Ho et al., 2020) specified in continuous time (Tzen & Raginsky, 2019; Song et al., 2021b; Chen et al., 2021; Kingma et al., 2021) is a generative model with latents $\mathbf{z} = \{\mathbf{z}_t \mid t \in [0, 1]\}$ obeying a forward process $q(\mathbf{z}|\mathbf{x})$ starting at data $\mathbf{x} \sim p(\mathbf{x})$. The forward process is a Gaussian process that satisfies the Markovian structure:

$$q(\mathbf{z}_t|\mathbf{x}) = \mathcal{N}(\mathbf{z}_t; \alpha_t \mathbf{x}, \sigma_t^2 \mathbf{I}), \quad q(\mathbf{z}_t|\mathbf{z}_s) = \mathcal{N}(\mathbf{z}_t; (\alpha_t/\alpha_s)\mathbf{z}_s, \sigma_{t|s}^2 \mathbf{I}) \tag{1}$$

where $0 \le s < t \le 1$, $\sigma_{t|s}^2 = (1 - e^{\lambda_t - \lambda_s})\sigma_t^2$, and $\alpha_t, \sigma_t$ specify a differentiable noise schedule whose log signal-to-noise-ratio $\lambda_t = \log[\alpha_t^2/\sigma_t^2]$ decreases with $t$ until $q(\mathbf{z}_1) \approx \mathcal{N}(\mathbf{0}, \mathbf{I})$. Learning to reverse the forward process for generation can be reduced to learning to denoise $\mathbf{z}_t \sim q(\mathbf{z}_t|\mathbf{x})$ into an estimate $\hat{\mathbf{x}}_\theta(\mathbf{z}_t, \lambda_t) \approx \mathbf{x}$ for all $t$ (we will drop the dependence on $\lambda_t$ to keep our notation clean).

---

[*]Equal contribution

We train this denoising model $\hat{\mathbf{x}}_\theta$ using a weighted mean squared error loss

$$\mathbb{E}_{\boldsymbol{\epsilon},t}\left[w(\lambda_t)\|\hat{\mathbf{x}}_\theta(\mathbf{z}_t) - \mathbf{x}\|_2^2\right] \tag{2}$$

over uniformly sampled times $t \in [0,1]$. This reduction of generation to denoising can be justified as optimizing a weighted variational lower bound on the data log likelihood under the diffusion model, or as a form of denoising score matching (Vincent, 2011; Song & Ermon, 2019; Ho et al., 2020; Kingma et al., 2021). In practice, we use the $\boldsymbol{\epsilon}$-prediction parameterization, defined as $\hat{\mathbf{x}}_\theta(\mathbf{z}_t) = (\mathbf{z}_t - \sigma_t \boldsymbol{\epsilon}_\theta(\mathbf{z}_t))/\alpha_t$, and train $\boldsymbol{\epsilon}_\theta$ using a mean squared error in $\boldsymbol{\epsilon}$ space with $t$ sampled according to a cosine schedule (Nichol & Dhariwal, 2021). This corresponds to a particular weighting $w(\lambda_t)$ for learning a scaled score estimate $\boldsymbol{\epsilon}_\theta(\mathbf{z}_t) \approx -\sigma_t \nabla_{\mathbf{z}_t} \log p(\mathbf{z}_t)$, where $p(\mathbf{z}_t)$ is the true density of $\mathbf{z}_t$ under $\mathbf{x} \sim p(\mathbf{x})$ (Ho et al., 2020; Kingma et al., 2021; Song et al., 2021b). We can sample from the model using the discrete time ancestral sampler, with classifier-free guidance when given a text conditioning signal (Appendix B).

## 3 VIDEO DIFFUSION MODELS

The standard architecture for $\hat{\mathbf{x}}_\theta$ in an image diffusion model is a U-Net (Ronneberger et al., 2015; Salimans et al., 2017), which is a neural net architecture constructed as a spatial downsampling pass followed by a spatial upsampling pass with skip connections to the downsampling pass activations. The network is built from layers of 2D convolutional residual blocks, for example in the style of the Wide ResNet (Zagoruyko & Komodakis, 2016), and each such convolutional block is followed by a spatial attention block (Vaswani et al., 2017; Wang et al., 2018; Chen et al., 2018).

We propose to extend this image diffusion model architecture to video data, given by a block of a fixed number of frames, using a particular type of 3D U-Net (Çiçek et al., 2016) that is factorized over space and time. First, we modify the image model architecture by changing each 2D convolution into a space-only 3D convolution, for instance, we change each 3x3 convolution into a 1x3x3 convolution (the first axis indexes video frames, the second and third index spatial height and width). The attention in each spatial attention block remains as attention over space; i.e., the first axis is treated as a batch axis. Second, after each spatial attention block, we insert a temporal attention block that performs attention over the first axis, treating the spatial axes as batch axes. We use relative position embeddings (Shaw et al., 2018) in each temporal attention block so that the network can distinguish ordering of frames in a way that does not require an absolute notion of video time. We visualize the model architecture in Appendix A.

The use of factorized space-time attention is known to be a good choice in video transformers for its computational efficiency (Arnab et al., 2021; Bertasius et al., 2021; Ho et al., 2019). An advantage of our factorized space-time architecture, which is unique to our video generation setting, is that it is particularly straightforward to mask the model to run on independent images rather than a video, simply by removing the attention operation inside each time attention block and fixing the attention matrix to exactly match each key and query vector at each video timestep. The utility of doing so is that it allows us to jointly train the model on both video and image generation. We find in our experiments that this joint training is important for sample quality (Section 4).

### 3.1 A NEW GRADIENT METHOD FOR CONDITIONAL GENERATION

The videos we consider modeling typically consist of hundreds to thousands of frames, at a frame rate of at least 24 frames per second. To manage the computational requirements of training our models, we only train on a small subset of say 16 frames at a time. However, at test time we can generate longer videos by extending our samples. For example, we could first generate a video $\mathbf{x}^a \sim p_\theta(\mathbf{x})$ consisting of 16 frames, and then extend it with a second sample $\mathbf{x}^b \sim p_\theta(\mathbf{x}^b|\mathbf{x}^a)$. If $\mathbf{x}^b$ consists of frames following $\mathbf{x}^a$, this allows us to autoregressively extend our sampled videos to arbitrary lengths, which we demonstrate in Section 4.3. Alternatively, we could choose $\mathbf{x}^a$ to represent a video of lower frame rate, and then define $\mathbf{x}^b$ to be those frames in between the frames of $\mathbf{x}^a$. This allows one to then to upsample a video temporally, similar to how Menick & Kalchbrenner (2019) generate high resolution images through spatial upsampling. Both approaches require one to sample from a conditional model, $p_\theta(\mathbf{x}^b|\mathbf{x}^a)$. This conditional model could be trained explicitly, but it can also be derived approximately from our unconditional model $p_\theta(\mathbf{x})$ by imputation, which has the advantage of not requiring a separately trained model. For example, Song et al. (2021b) present a general method for conditional sampling from a jointly trained diffusion model $p_\theta(\mathbf{x} = [\mathbf{x}^a, \mathbf{x}^b])$. In their approach to sampling from $p_\theta(\mathbf{x}^b|\mathbf{x}^a)$, the sampling procedure for updating $\mathbf{z}_s^b$ is unchanged from the

standard method for sampling from $p_\theta(\mathbf{z}_s | \mathbf{z}_t)$, with $\mathbf{z}_s = [\mathbf{z}_s^a, \mathbf{z}_s^b]$, but the samples for $\mathbf{z}_s^a$ are replaced by exact samples from the forward process, $q(\mathbf{z}_s^a | \mathbf{x}^a)$, at each iteration. The samples $\mathbf{z}_s^a$ then have the correct marginal distribution by construction, and the samples $\mathbf{z}_s^b$ will conform with $\mathbf{z}_s^a$ through their effect on the denoising model $\hat{\mathbf{x}}_\theta([\mathbf{z}_t^a, \mathbf{z}_t^b])$. Similarly, we could sample $\mathbf{z}_s^a$ from $q(\mathbf{z}_s^a | \mathbf{x}^a, \mathbf{z}_t^a)$, which follows the correct conditional distribution in addition to the correct marginal. We will refer to both of these approaches as the *replacement* method for conditional sampling from diffusion models.

When we tried the replacement method to conditional sampling, we found it to not work well for our video models: Although samples $\mathbf{x}^b$ looked good in isolation, they were often not coherent with $\mathbf{x}^a$. This is caused by a fundamental problem with this replacement sampling method. That is, the latents $\mathbf{z}_s^b$ are updated in the direction provided by $\hat{\mathbf{x}}_\theta^b(\mathbf{z}_t) \approx \mathbb{E}_q[\mathbf{x}^b | \mathbf{z}_t]$, while what's needed instead is $\mathbb{E}_q[\mathbf{x}^b | \mathbf{z}_t, \mathbf{x}^a]$. Writing this in terms of the score of the data distribution, we get $\mathbb{E}_q[\mathbf{x}^b | \mathbf{z}_t, \mathbf{x}^a] = \mathbb{E}_q[\mathbf{x}^b | \mathbf{z}_t] + (\sigma_t^2/\alpha_t) \nabla_{\mathbf{z}_t^b} \log q(\mathbf{x}^a | \mathbf{z}_t)$, where the second term is missing in the replacement method. Assuming a perfect denoising model, plugging in this missing term would make conditional sampling exact. Since $q(\mathbf{x}^a | \mathbf{z}_t)$ is not available in closed form, however, we instead propose to approximate it using a Gaussian of the form $q(\mathbf{x}^a | \mathbf{z}_t) \approx \mathcal{N}[\hat{\mathbf{x}}_\theta^a(\mathbf{z}_t), \sigma_t^2 \mathbf{I}]$. Assuming a perfect model, this approximation becomes exact as $t \to 0$, and empirically we find it to be good for larger $t$ also. Plugging in the approximation, our proposed method to conditional sampling is a variant of the replacement method with an adjusted denoising model, $\tilde{\mathbf{x}}_\theta^b$, defined by

$$\tilde{\mathbf{x}}_\theta^b(\mathbf{z}_t) = \hat{\mathbf{x}}_\theta^b(\mathbf{z}_t) - \frac{1}{2\alpha_t} \nabla_{\mathbf{z}_t^b} \|\mathbf{x}^a - \hat{\mathbf{x}}_\theta^a(\mathbf{z}_t)\|_2^2 . \tag{3}$$

We refer to this as the *gradient method* to conditional sampling. The method also extends to the case of spatial interpolation (or super-resolution), in which the mean squared error loss is imposed on a downsampled version of the model prediction, and backpropagation is performed through this downsampling (see Appendix C.3). We empirically investigate this method in Section 4.3.

## 4 EXPERIMENTS

We evaluate our models on unconditional and text-conditioned video generation. For text-conditioned video generation, we train on a dataset of 10 million captioned videos with a spatial resolution of 64x64 pixels, and we condition the diffusion model on captions in the form of BERT-large embeddings (Devlin et al., 2019) processed using attention pooling. For unconditional generation, we train and evaluate models on an existing benchmark (Soomro et al., 2012).

### 4.1 JOINT TRAINING ON VIDEO AND IMAGE MODELING

As described in Section 3, one of the main advantages of our video architecture is that it allows us to easily train the model jointly on video and image generative modeling objectives. To implement this joint training, we concatenate random independent image frames to the end of each video sampled from the dataset, and we mask the attention in the temporal attention blocks to prevent mixing information across video frames and each individual image frame. We choose these random independent images from random videos within the same dataset; in future work we plan to explore the effect of choosing images from other larger image-only datasets. In Table 1, one can see clear improvements in video and image sample quality metrics as more independent image frames are added. Adding independent image frames has the effect of reducing variance of the gradient at the expense of some bias for the video modeling objective, and thus it can be seen as a memory optimization to fit more independent examples in a batch.

| Image frames | FVD↓ | FID-avg↓ | IS-avg↑ | FID-first↓ | IS-first↑ |
|---|---|---|---|---|---|
| 0 | 202.28/205.42 | 37.52/37.40 | 7.91/7.58 | 41.14/40.87 | 9.23/8.74 |
| 4 | 68.11/70.74 | 18.62/18.42 | 9.02/8.53 | 22.54/22.19 | 10.58/9.91 |
| 8 | 57.84/60.72 | 15.57/15.44 | 9.32/8.82 | 19.25/18.98 | 10.81/10.12 |

Table 1: Improved sample quality due to image-video joint training on text-to-video generation (small model). The videos are 16x64x64, and we consider training on an additional 0, 4, or 8 independent image frames per video. Metrics are reported on 4096 samples. FVD is a video metric; FID/IS are image metrics, which we measure by averaging activations across frames (FID/IS-avg) and by measuring the first frame only (FID/IS-first). For FID and FVD, the two listed numbers are measured against the training and validation sets, respectively. For IS, the two listed numbers are averaged scores across 1 split and 10 splits of samples, respectively.

## 4.2 EFFECT OF CLASSIFIER-FREE GUIDANCE

Table 2 reports results that verify the effectiveness of classifier-free guidance (Ho & Salimans, 2021) on text-to-video generation. As expected, there is clear improvement in the Inception Score-like metrics with higher guidance weight, while the FID-like metrics improve and then degrade with increasing guidance weight. Similar findings have been reported on text-to-image generation (Nichol et al., 2021). See Appendix C.1 for samples.

| Frameskip | Guidance weight | FVD↓ | FID-avg↓ | IS-avg↑ | FID-first↓ | IS-first↑ |
|---|---|---|---|---|---|---|
| 1 | 1.0 | 41.65/43.70 | 12.49/12.39 | 10.80/10.07 | 16.42/16.19 | 12.17/11.22 |
|  | 2.0 | 50.19/48.79 | 10.53/10.47 | 13.22/12.10 | 13.91/13.75 | 14.81/13.46 |
|  | 5.0 | 163.74/160.21 | 13.54/13.52 | 14.80/13.46 | 17.07/16.95 | 16.40/14.75 |
| 4 | 1.0 | 56.71/60.30 | 11.03/10.93 | 9.40/8.90 | 16.21/15.96 | 11.39/10.61 |
|  | 2.0 | 54.28/51.95 | 9.39/9.36 | 11.53/10.75 | 14.21/14.04 | 13.81/12.63 |
|  | 5.0 | 185.89/176.82 | 11.82/11.78 | 13.73/12.59 | 16.59/16.44 | 16.24/14.62 |

Table 2: Effect of classifier-free guidance on text-to-video generation (large models). Sample quality is reported for 16x64x64 models trained on frameskip 1 and 4 data. The model was jointly trained on 8 independent image frames per 16-frame video.

## 4.3 AUTOREGRESSIVE VIDEO EXTENSION FOR LONGER SEQUENCES

In Section 3.1 we proposed the *gradient method* for conditional sampling from diffusion models, an improvement over the *replacement method* of Song et al. (2021b). In Table 3 we present results on generating longer videos using both techniques, and find that our proposed method indeed improves over the replacement method in terms of perceptual quality scores. See Appendix C.2 for samples.

| Guidance weight | Conditioning method | FVD↓ | FID-avg↓ | IS-avg↑ | FID-first↓ | IS-first↑ |
|---|---|---|---|---|---|---|
| 2.0 | gradient | 136.22/134.55 | 13.77/13.62 | 10.30/9.66 | 16.34/16.46 | 14.67/13.37 |
|  | replacement | 451.45/436.16 | 25.95/25.52 | 7.00/6.75 | 16.33/16.46 | 14.67/13.34 |
| 5.0 | gradient | 133.92/133.04 | 13.59/13.58 | 10.31/9.65 | 16.28/16.53 | 15.09/13.72 |
|  | replacement | 456.24/441.93 | 26.05/25.69 | 7.04/6.78 | 16.30/16.54 | 15.11/13.69 |

Table 3: Generating 64x64x64 videos using autoregressive extension of 16x64x64 models.

## 4.4 UNCONDITIONAL VIDEO MODELING

To compare our approach with existing methods in the literature, we use a popular benchmark of Soomro et al. (2012) for unconditional modeling of video. The benchmark consists of short clips of people performing one of 101 activities, and was originally collected for the purpose of training action recognition models. We model short segments of 16 frames from this dataset, downsampled to a spatial resolution of 64x64. In Table 4 we present perceptual quality scores for videos generated by our model, and we compare against methods from the literature, finding that our method strongly improves upon the previous state-of-the-art. We discuss evaluation metrics in Appendix D.

| Method | resolution | FID↓ | IS↑ |
|---|---|---|---|
| MoCoGAN (Tulyakov et al., 2018) | 16x64x64 | $26998 \pm 33$ | 12.42 |
| TGAN-F (Kahembwe & Ramamoorthy, 2020) | 16x64x64 | $8942.63 \pm 3.72$ | 13.62 |
| TGAN-ODE (Gordon & Parde, 2021) | 16x64x64 | $26512 \pm 27$ | 15.2 |
| TGAN-F (Kahembwe & Ramamoorthy, 2020) | 16x128x128 | $7817 \pm 10$ | $22.91 \pm .19$ |
| VideoGPT (Yan et al., 2021) | 16x128x128 |  | $24.69 \pm 0.30$ |
| TGAN-v2 (Saito et al., 2020) | 16x64x64 | $3431 \pm 19$ | $26.60 \pm 0.47$ |
| TGAN-v2 (Saito et al., 2020) | 16x128x128 | $3497 \pm 26$ | $28.87 \pm 0.47$ |
| DVD-GAN (Clark et al., 2019) | 16x128x128 |  | $32.97 \pm 1.7$ |
| **ours** | 16x64x64 | **295** | **57.1** |
| real data | 16x64x64 |  | 60.2 |

Table 4: Unconditional generative modeling of UCF101 video data. See Appendix D for discussion.

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

## A DENOISING MODEL ARCHITECTURE

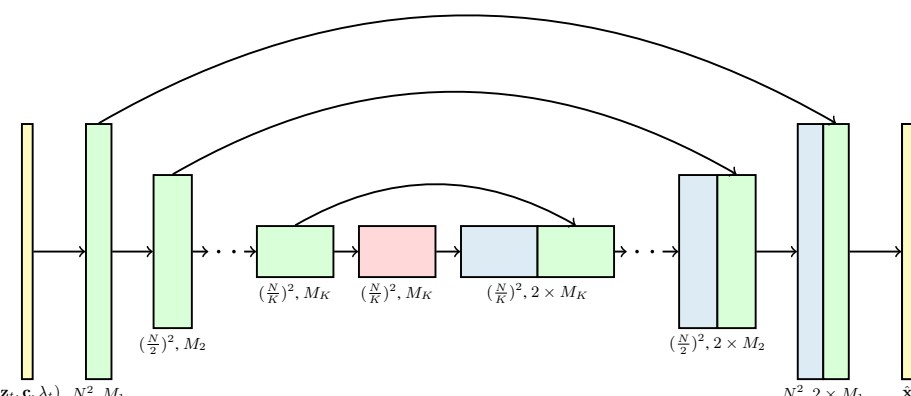

Figure 1: The U-Net architecture for $\hat{\mathbf{x}}_\theta$ in the diffusion model. Each block represents a 4D tensor with axes labeled as frames × height × width × channels. The input is a noisy video $\mathbf{z}_t$, conditioning $\mathbf{c}$, and the log SNR $\lambda_t$. The downsampling/upsampling blocks adjust the spatial input resolution height × width by a factor of 2 through each of the $K$ blocks. The channel counts are specified using channel multipliers $M_1, M_2, ..., M_K$, and the upsampling pass has concatenation skip connections to the downsampling pass.

## B SAMPLING FROM DIFFUSION MODELS

To sample from the model, we use the discrete time ancestral sampler (Ho et al., 2020). To define this sampler, first note that the forward process can be described in reverse as $q(\mathbf{z}_s|\mathbf{z}_t, \mathbf{x}) = \mathcal{N}(\mathbf{z}_s; \tilde{\boldsymbol{\mu}}_{s|t}(\mathbf{z}_t, \mathbf{x}), \tilde{\sigma}^2_{s|t}\mathbf{I})$ (noting $s < t$), where $\tilde{\boldsymbol{\mu}}_{s|t}(\mathbf{z}_t, \mathbf{x}) = e^{\lambda_t - \lambda_s}(\alpha_s/\alpha_t)\mathbf{z}_t + (1 - e^{\lambda_t - \lambda_s})\alpha_s\mathbf{x}$ and $\tilde{\sigma}^2_{s|t} = (1 - e^{\lambda_t - \lambda_s})\sigma^2_s$. Starting at $\mathbf{z}_1 \sim \mathcal{N}(\mathbf{0}, \mathbf{I})$, the ancestral sampler follows the rule

$$\mathbf{z}_s = \tilde{\boldsymbol{\mu}}_{s|t}(\mathbf{z}_t, \hat{\mathbf{x}}_\theta(\mathbf{z}_t)) + \sqrt{(\tilde{\sigma}^2_{s|t})^{1-\gamma}(\sigma^2_{t|s})^\gamma}\boldsymbol{\epsilon} \quad (4)$$

where $\boldsymbol{\epsilon}$ is standard Gaussian noise, $\gamma$ is a hyperparameter that controls the stochasticity of the sampler (Nichol & Dhariwal, 2021), and $s, t$ follow a uniformly spaced sequence from 1 to 0. Other sampling algorithms such as DDIM (Song et al., 2021a) can be used as well.

In the conditional generation setting, the data $\mathbf{x}$ is equipped with a conditioning signal $\mathbf{c}$, which may represent a class label, text caption, or other type of conditioning. To train a diffusion model to fit $p(\mathbf{x}|\mathbf{c})$, the only modification that needs to be made is to provide $\mathbf{c}$ to the model as $\hat{\mathbf{x}}_\theta(\mathbf{z}_t, \mathbf{c})$. Improvements to sample quality can be obtained in this setting by using *classifier-free guidance* (Ho & Salimans, 2021). This method samples using adjusted model predictions $\tilde{\boldsymbol{\epsilon}}_\theta$, constructed via

$$\tilde{\boldsymbol{\epsilon}}_\theta(\mathbf{z}_t, \mathbf{c}) = (1 + w)\boldsymbol{\epsilon}_\theta(\mathbf{z}_t, \mathbf{c}) - w\boldsymbol{\epsilon}_\theta(\mathbf{z}_t), \quad (5)$$

where $w$ is the *guidance strength*, $\boldsymbol{\epsilon}_\theta(\mathbf{z}_t, \mathbf{c}) = \frac{1}{\sigma_t}(\mathbf{z}_t - \hat{\mathbf{x}}_\theta(\mathbf{z}_t, \mathbf{c}))$ is the regular conditional model prediction, and $\boldsymbol{\epsilon}_\theta(\mathbf{z}_t)$ is a prediction from an unconditional model jointly trained with the conditional model (if $\mathbf{c}$ consists of embedding vectors, unconditional modeling can be represented as $\mathbf{c} = \mathbf{0}$). For $w > 0$ this adjustment has the effect of over-emphasizing the effect of conditioning on the signal $\mathbf{c}$, which tends to produce samples of lower diversity but higher quality compared to sampling from the regular conditional model (Ho & Salimans, 2021). The method can be interpreted as a way to guide the samples towards areas where an implicit classifier $p(\mathbf{c}|\mathbf{z}_t)$ has high likelihood, and is an adaptation of the explicit classifier guidance method proposed by Dhariwal & Nichol (2021).

## C TEXT-CONDITIONAL VIDEO MODELING

Video samples accompanying the following material are provided at `https://video-diffusion.github.io/`.

## C.1 Effect of classifier-free guidance

Figure 2 shows the effect of classifier-free guidance (Ho & Salimans, 2021) on a text-conditioned video model. Similar to what was observed in other work that used classifier-free guidance on text-conditioned image generation (Nichol et al., 2021) and class-conditioned image generation (Ho & Salimans, 2021; Dhariwal & Nichol, 2021), adding guidance increases the sample fidelity of each individual image and emphases the effect of the conditioning signal.

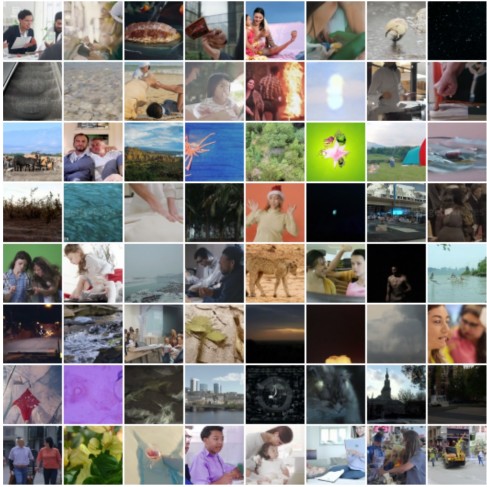 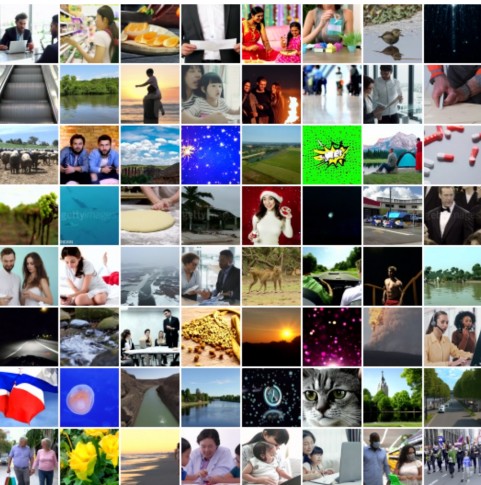

Figure 2: Example frames from a random selection of videos generated by our 16x64x64 text-conditioned model. Left: unguided samples, right: guided samples using classifier-free guidance.

## C.2 Effect of replacement vs gradient

Figure 3 shows the samples of our gradient method for conditional sampling compared to the replacement method (Section 3.1) for the purposes of generating long samples in a block-autoregressive manner (Section 4.3). The samples from the replacement method clearly show a lack of temporal coherence, since frames from different blocks throughout the generated videos appear to be uncorrelated samples (conditioned on $\mathbf{c}$). The samples from the gradient method, by contrast, are clearly temporally coherent over the course of the entire autoregressive generation process.

## C.3 Gradient method for super-resolution

As mentioned in Section 3.1, we also use our gradient method to perform super-resolution using an unconditional high resolution diffusion model. In this setting, we have low resolution ground truth videos $\mathbf{x}^a$ (e.g. at the 64x64 spatial resolution), which may be generated from a low resolution model, and we wish to upsample them into high resolution videos (e.g. at the 128x128 spatial resolution) using an unconditional high resolution diffusion model $\hat{\mathbf{x}}_\theta$. To accomplish this, we adjust the high resolution model as follows:

$$\tilde{\mathbf{x}}_\theta(\mathbf{z}_t) = \hat{\mathbf{x}}_\theta(\mathbf{z}_t) - \frac{1}{2\alpha_t}\nabla_{\mathbf{z}_t}\|\mathbf{x}^a - \hat{\mathbf{x}}_\theta^a(\mathbf{z}_t)\|_2^2 \tag{6}$$

where $\hat{\mathbf{x}}_\theta^a$ is defined to be the high resolution model output downsampled using a differentiable downsampling algorithm such as bilinear interpolation.

Note that it is possible to simultaneously condition on low resolution videos while autoregressively extending samples at the high resolution, both using the gradient method. In Fig. 4, we show samples of this approach for extending 16x64x64 low resolution samples at frameskip 4 to 64x128x128 samples at frameskip 1 using a 9x128x128 diffusion model.

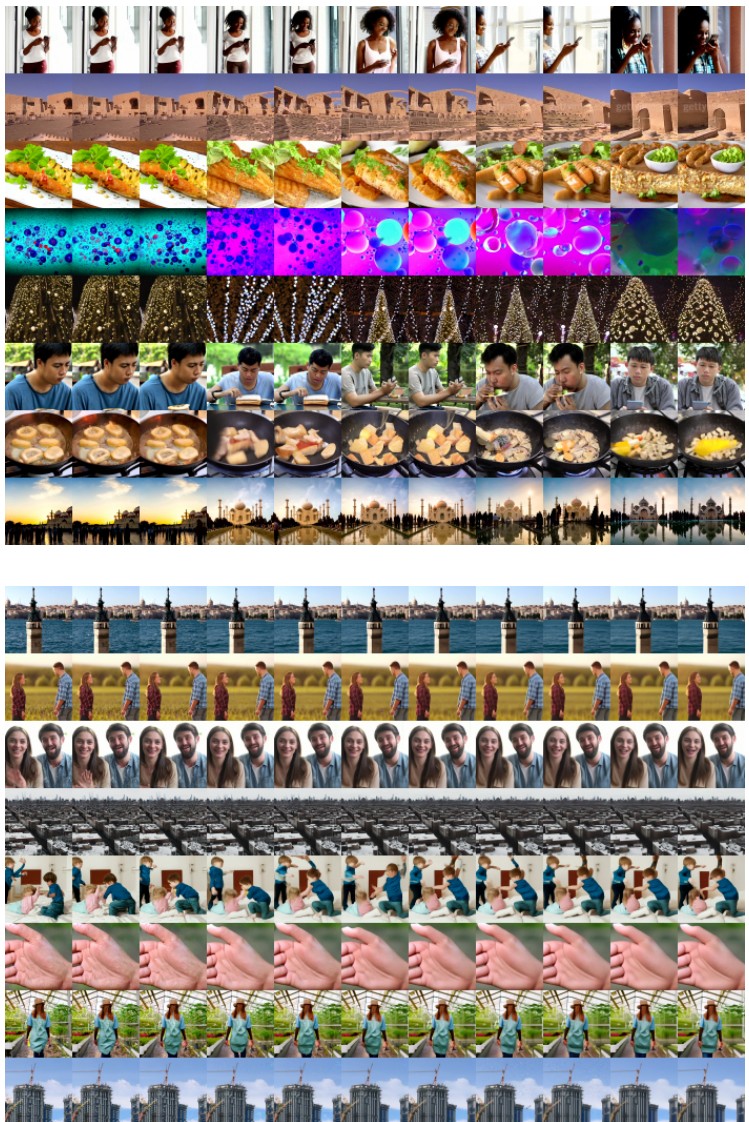

Figure 3: Comparing the replacement method (top) vs the gradient method (bottom) for conditioning for block-autoregressive generation of 64 frames from a 16 frame model. Video frames are displayed over time from left to right; each row is an independent sample. The replacement method suffers from a lack of temporal coherence, unlike the gradient method.

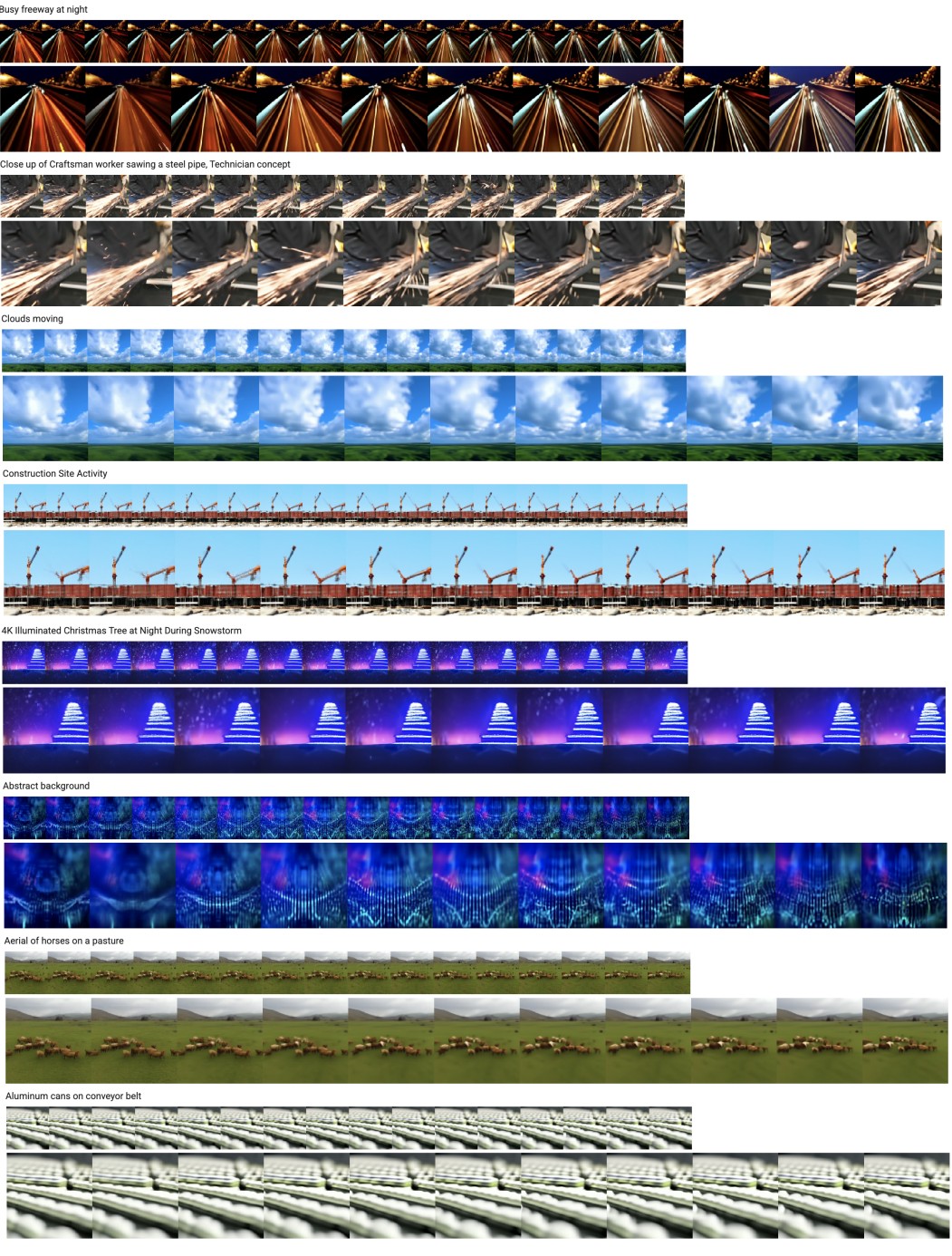

Figure 4: Text-conditioned video samples from a cascade of two models. First samples are generated from a 16x64x64 frameskip 4 model. Then those samples are treated as ground truth for simultaneous super-resolution and autoregressive extension to 64x128x128 using a 9x128x128 frameskip 1 model. Both models are conditioned on the text prompt. In this figure, the text prompt, low resolution frames, and high resolution frames are visualized in sequence.

## D  UNCONDITIONAL VIDEO MODELING

Similar to previous methods on this benchmark (Soomro et al., 2012), we use the C3D model (Tran et al., 2015) as implemented at `github.com/pfnet-research/tgan2` (Saito et al., 2020) for calculating FID and IS, using 10,000 samples generated from our model. The C3D model internally resizes the input data to 112x112 resolution, so perceptual scores are approximately comparable even when the data is sampled at a different resolution originally. Furthermore, as discussed by Yushchenko et al. (2019), methods in the literature are unfortunately not always consistent in the data preprocessing that is used, which may lead to small differences in reported scores between papers. We use the data loader provided by TensorFlow Datasets (TFD, 2022) without further processing, and we train on all 13,320 videos. The Inception Score we calculate for real data ($\approx 60$) is consistent with that reported by Kahembwe & Ramamoorthy (2020). They report a higher real data inception score of $\approx 90$ for data sampled at the 128x128 resolution, which indicates that our 64x64 model might be at a disadvantage compared to works that generate at that resolution. Nevertheless, our model obtains the best perceptual quality metrics that we could find in the literature.

