# OpenReview forum: "Video Diffusion Models"
_ICLR.cc/2022/Workshop/DGM4HSD — ICLR 2022 DGM4HSD workshop Poster_

### Official Review · Reviewer_gJ4L · 2022-03-23
**Good proof of concept of diffusion models for video**

**Rating:** 7
**Confidence:** 3

**Review:**

The authors present a diffusion model for video. Building on the established approach to image generation with diffusion models, they first present a generative model for fixed-length videos, before discussing a block-autoregressive extension to variable-length video generation. The authors present experiments on unconditional and text-conditional video generation.

The work is reasonably novel and tackles an interesting problem with a method that is increasingly receiving attention. As a generative model for video data, it is also quite relevant for the workshop.

While I am neither a diffusion expert nor very familiar with the video generation literature, the proposed setup makes sense and the authors justify their choices well. The experiments are thorough and include both typical generative model experiments as well as a range of ablations of the proposed method components. They show that diffusion models can generate video data and support the choices made by the authors. I appreciated the sample frames in the appendix; if possible, additional sample video data in the supplementary material would be great.

In general, the paper is very well written. The authors manage to convey a good mix of a high-level summary of the diffusion approach, technical details of their method, as well as intuitions and explanations, especially considering the tight page limit.

Overall, I thank the authors for a good paper that I enjoyed reading. In my opinion, this should clearly be accepted at the workshop.

Minor comments / questions:
- Appendix B: In the ancestral sampler, how are the values for s (or t) chosen?
- Some discussion of the required compute and if / how this approach can be scaled to higher resolution would be great.

---

### Official Review · Reviewer_kzvT · 2022-03-23
**Good work, supplementary video results will be helpful**

**Rating:** 7
**Confidence:** 2

**Review:**

### Summary
This paper extends image diffusion models to support videos. Specifically, the authors build a 3D U-Net which they factorize between space and time. Specifically they factorize each 3x3 convolution into 1x3x3 convolution, and have separate blocks for spatial vs. temporal attention. Due to separation of concerns, the model is able to train with both video and image data. The resulting model is applicable to conditional as well as unconditional video generation applications. Finally, the authors provide two recipes for generating long videos using this model.


### Evaluation
Despite my limited expertise in this area, I believe that this video generation method is interesting and technically sound. It is also novel as far as I can tell.

It is hard to examine the quality of results without looking at any supplementary video results. I am particularly curious about temporal coherence of the output videos, as well as quality of content across long videos. My main concern with temporal stability stems from the separation of spatial and temporal factors. This can potentially limit spatial information flow across time, i.e. flow of moving objects. Some additional results would really help evaluate the impact of this choice.

Nevertheless, I think this work is interesting for this workshop and I support acceptance.

---

### Official Review · Reviewer_mg4r · 2022-03-27
**Good work**

**Rating:** 7
**Confidence:** 3

**Review:**

Authors’ contributions are threefold. First, they propose a neural architecture for a diffusion generative model for videos and evaluate it empirically on UCF101, one of the benchmark dataset for this task, yielding state-of-the-art performance on FID and IS metrics in unconditional video. Second, they propose a gradient method for conditional sampling in diffusion models, and show it’s improving upon the replacement method from earlier literature. Thirdly, they propose to add independently sampled frames/images to the batches of sequential video frames to reduce the variance of the gradient, and they show its significant impact on the results. Additionally, they also evaluate text-conditional video generation, but as a reader I find it difficult to interpret these results given only the set of metrics reported since none of them allows me to access how successful is this approach at making the content of the video actually relevant to the text prompt conditioned on. (more on this in detailed feedback)

To the best of my knowledge, all these contributions are novel & timely. The paper is well written, contributions are well motivated, described and explained. The subject is relevant to the theme of the workshop. In my opinion, an accept.

*More detailed:*
If I understand Appendix C.1 correctly (“adding guidance increases the sample ﬁdelity of each individual image”), I would add “Guidance weight=0.0” to Table 2, such that this takeaway can be seen in the main text, and I would narrate it in the main text as well, rather than only in the Appendix. I think that’s an interesting observation that I didn’t know about, and a valuable observation the same effect takes place for video diffusion models.

*Typo:*
>​​We can sample from the model using the discrete time ancestral sampler **or** with classiﬁer-free guidance.

---

### Decision · Program_Chairs · 2022-03-26

Accept (Poster)